# Sleep, Diet, and Exercise: How Much Dementia Caregivers Are Affected?

**DOI:** 10.3390/brainsci14080826

**Published:** 2024-08-17

**Authors:** Angeliki Tsapanou, Panagiota Zoi, Paraskeui Sakka

**Affiliations:** 1Athens Alzheimer’s Association, 11636 Athens, Greece; pzoi@alzheimerathens.gr (P.Z.); info@psakka.gr (P.S.); 2Cognitive Neuroscience Division, Columbia University, New York, NY 10032, USA

**Keywords:** sleep, diet, exercise, dementia, caregivers

## Abstract

The current descriptive study reports the sleep, diet, and exercise patterns among 114 dementia caregivers, whose mean age was 55.7 (SD: 10.4) years, with 83 (72.8%) being women. The results indicate significant sleep dysfunction: 37.2% of caregivers reported rarely or never feeling rested upon waking, and 46.5% did not get enough sleep, with 45.6% sleeping only 5 to 5.5 h on average. Sleep latency was also prevalent, as 33.3% required 16 to 30 min to fall asleep. Dietary habits showed reliance on coffee, with 69.4% consuming it daily. Meat consumption was reported by 75%, and 60.9% ate pasta, indicating common dietary preferences. While 86.2% had one to three meals per day, 100% of the caregivers supplemented their diets with vitamins. The physical activity level was low, with 62.3% of respondents reporting no exercise in the past week. These findings underscore significant health concerns among dementia caregivers, including sleep deprivation, inadequate nutrition, and physical inactivity. The report emphasizes the need for targeted interventions to promote self-care practices that can enhance caregivers’ health, including better sleep hygiene, balanced nutrition, and regular exercise.

## 1. Introduction

Caring for individuals with dementia is a challenging and rewarding endeavor that demands considerable mental, emotional, and physical resilience. As such, the well-being of caregivers themselves—encompassing sleep, diet, and exercise—is paramount, yet it is often overlooked in the broader discourse on caregiving. Ensuring adequate sleep, maintaining a balanced diet, and regular exercise are not just beneficial but essential components of a caregiver’s health regimen, playing a crucial role in their capacity to provide sustained and effective care.

Scientific research has provided valuable insights into the impact of sleep, exercise, and psychosocial interventions on the health and well-being of dementia caregivers. Both cohabiting and non-cohabiting caregivers of people with dementia experience poor sleep quality compared to non-caregivers [1]. Three major factors are also very crucial for the sleep function of the caregivers: the presence of a disrupted sleep routine, overall burden and depression, and physical health status [2]. A comprehensive systematic review and meta-analysis revealed that dementia caregivers often experience poorer sleep quality and shorter sleep durations compared to non-caregivers. However, it was found that noninvasive behavioral interventions, including sleep hygiene education, light chronotherapy, and relaxation techniques at bedtime, significantly improved caregivers’ sleep quality [3]. These findings underscore the bidirectional relationship between caregiving and sleep, where caregiving responsibilities can disrupt sleep, and poor sleep can in turn impair the caregiver’s ability to provide care. Importantly, the analysis highlighted that even though the caregiving environment is stressful, caregivers can still adopt behaviors that improve their sleep, suggesting that behavioral interventions are crucial for this population.

Interestingly, regarding diet habits, a qualitative study showed that dementia care helps family caregivers become more conscious of their own dietary lifestyle choices and increases communication opportunities. On the other hand, caregiver burden was found to have a negative influence on the quality and level of interest in dietary choices of caregivers [4]. However, a study analyzing micronutrient recommended daily allowances showed that caregivers may lack proper nutrients, and care hours may influence their health habits [5].

Regarding physical health, a home-based exercise program of two 60 min sessions per week significantly improved the quality of life of informal female caregivers of people with dementia [6]. Furthermore, another study focused on the effectiveness of a home-based exercise intervention for caregivers, specifically utilizing sitting Tai Chi, highlighted the potential benefits of exercise [7]. This intervention involved the caregivers participating in a series of sitting Tai Chi sessions, which are known to be moderate-intensity aerobic exercises. The study aimed to evaluate the impact of these sessions on caregivers’ mental and physical health, suggesting the importance of physical activity for reducing depressive symptoms and improving overall well-being among caregivers. Furthermore, as they are often examined secondarily, caregivers are overlooked for participation in interventions with care recipients. Caregivers seem to significantly improve both their psychosocial and physical health when exercising together with care recipients [8].

The aforementioned findings collectively illustrate the critical importance of targeted studies to support dementia caregivers, addressing the complexity of sleep, exercise, and diet needs and habits. Through adopting evidence-based strategies, caregivers can significantly improve their own health and resilience, ultimately enhancing the quality of care they provide to their loved ones with dementia. In the current descriptive study, we report the sleep, diet, and exercise habits of dementia caregivers in Greece.

## 2. Methods

Participants were recruited from the Athens Alzheimer’s Association (https://alzheimerathens.gr/en/about-us-2/) (accessed on April 15, 2024). All participants were caregivers of people with all kinds of dementias, of all stages. The study primarily involved familial caregivers rather than professional ones, with participants drawn from the Athens Alzheimer’s Association. These participants were either actively caring for a person with dementia or engaged with the Association’s services, such as attending a dementia information seminar. Data collection was conducted solely through in-person methods, ensuring no online evaluations were carried out. Participation in the study was entirely voluntary. Data collection took place between February and March 2024. The study was approved by the Institutional Review Committee (IRB) of the Athens Alzheimer’s Association (#4/15 February 2024). All participants have a signed consent form.

The following self-rated questionnaires were provided to the caregivers, regarding their own health:

### 2.1. Sleep

The Medical Outcomes Study-Sleep Scale was used [9]. The scale uses predominantly Likert-type questions to evaluate sleep. The scale range from 1 (meaning “all of the time”) to 6 (“none of the time”), and require respondents to indicate how frequently during the previous 4 weeks they have experienced certain sleep-related issues. Several of these items are reverse scored.

### 2.2. Diet

The Food Frequency Questionnaire was used [10]. This is a checklist of foods and beverages with a frequency response section for subjects to report how often each item was consumed over a specified period of time.

### 2.3. Exercise

The Historical Adulthood Physical Activity Questionnaire (hapaq) [11]. This is an extended questionnaire regarding the daily physical activity patterns, the activity at work, and other recreational physical activities the person might have engaged in during the last 4 weeks.

Questionnaires were provided in Greek. Further information regarding the questionnaires and their previous use in a Greek sample can be found in a previous study [12]. For the purposes of the current analyses, all answers were trichotomized.

## 3. Results

A total of 114 caregivers filled out the questionnaires for sleep, diet, and exercise. The mean age of the participants was 55.7 years old, with 14.2 mean years of education (see Table 1). The majority of the caregivers were women (72.8%).

### 3.1. Sleep

The majority of the caregivers (33.3%) reported needing a mean of 16–30 min to fall asleep, while their mean hours of sleep was 5–5.30 (45.6%).

The questions for which a sleep dysfunction was vastly reported were:

“Get enough sleep to feel rested upon waking in the morning”, which was reported “a little of the time” or “none of the time” by 37.2% of the participants (see Figure 1). Similarly, the majority of the caregivers (46.5%) reported “not getting the amount of sleep needed”.

### 3.2. Diet

The vast majority of the caregivers (69.4%) reported drinking coffee one, two, or more than two times a day. One to six times per week was reported eating meat’s fat (75%), and having any kind of pasta for 60.9% of the participants.

The majority of the participants (86.2%) reported having 1–3 meals per day, including snacks. Interestingly, all caregivers (100%) reported taking vitamins.

### 3.3. Exercise

Regarding exercise we mainly focused on the following question: “During the past 7 days did you exercise?”, with 62.3% of the caregivers reporting not having exercised (see Figure 2).

## 4. Discussion

Caregivers of people with dementia face unique challenges and stressors that can have significant impacts on their physical and mental health, on top of the burden they carry from the disease. Caregiving for someone with dementia can be emotionally and psychologically draining, leading to increased risks of stress, depression, and anxiety. Healthy habits, including regular physical activity, can improve mental health and resilience, helping caregivers manage stress and maintain a positive outlook.

The survey results indicate significant sleep dysfunction among the caregivers. A notable 37.2% reported rarely or never feeling rested upon waking, while 46.5% of caregivers acknowledged not obtaining the necessary amount of sleep. These findings highlight prevalent sleep issues within this group, likely influenced by the stresses of caregiving responsibilities. Moreover, almost half of the respondents (45.6%) reported that they sleep for between 5 and 5.5 h a night on average, well below the 7–8 h typically recommended for adults.

Additionally, with a third of respondents taking 16 to 30 min to fall asleep, there may be evidence of sleep latency issues, further affecting overall sleep quality.

The dietary habits of the caregivers offer insights into their nutrition and lifestyle. A significant majority (69.4%) of caregivers consume coffee one or more times daily, which may suggest a reliance on caffeine to combat fatigue and tiredness. The prevalence of meat and pasta consumption—75% and 60.9% of participants, respectively—shows typical dietary choices, which may impact overall health if not balanced. The majority reported consuming one to three meals a day, including snacks, and all participants claimed to take vitamins, potentially in an attempt to supplement their diet.

The survey revealed a concerning lack of exercise among caregivers, with 62.3% reporting no exercise in the past week. This lack of physical activity could exacerbate the sleep difficulties and health issues found among this group, suggesting a need for interventions aimed at promoting healthier lifestyle habits.

When studying the health behaviors and outcomes of caregivers, it is essential to consider a range of potential confounding variables that could significantly influence the results. Socioeconomic status, for example, affects access to healthcare resources, the quality of nutrition, and overall living conditions, which in turn can impact a caregiver’s ability to maintain their health while managing caregiving duties [13]. Additionally, pre-existing health conditions among caregivers are critical to consider as they may predispose individuals to different stress-related illnesses or amplify the health impacts of caregiving [14,15]. Although our sample was quite diverse, these factors should be incorporated into further research to control for possible factors affecting the association between the disease and its effect on the caregivers.

Caregivers often experience disrupted sleep patterns due to the nighttime needs of those they care for or from stress-induced insomnia [2]. In terms of diet, caregivers may find themselves opting for more convenient, less nutritious food choices because of time constraints, leading to poorer overall health [14]. Additionally, the physical and emotional exhaustion associated with caregiving responsibilities can diminish both the time available for exercise and the motivation to engage in physical activity [16]. These lifestyle changes, prompted by the demands of caregiving, can significantly deteriorate a caregiver’s health. Cultural factors significantly influence caregiving practices and health behaviors, particularly in a culturally distinct setting like Greece, where familial obligations and strong intergenerational support are deeply ingrained. These norms often dictate that caregiving responsibilities fall primarily to family members, influencing both the extent and nature of care provided. The traditional Mediterranean diet prevalent in Greece may generally guide caregivers towards healthier nutritional choices, yet the stress of caregiving could lead to deviations such as increased consumption of less healthy comfort foods [17]. Furthermore, the cultural emphasis on familial care may result in underutilization of formal support services, increasing the physical and emotional burdens on caregivers, which can negatively impact their physical activity and sleep quality [18]. Understanding these cultural nuances is crucial, suggesting that support interventions in Greece need to be culturally tailored to leverage community and familial support while enhancing access to and acceptance of external caregiving resources.

Greece has implemented some policies and strategies to deal with dementia and support dementia caregivers. Specifically, the National Strategy for Dementia in Greece is the National Action Plan for Dementia and Alzheimer’s Disease launched by the Ministry of Health, it focuses on improving the quality of life for people with dementia and their caregivers. This includes early diagnosis, comprehensive care services, and public awareness. Further, The Greek Dementia Helpline 1102 as a telephone service staffed by psychologists, neurologists, and social workers with expertise in dementia care and is provided by the Athens Alzheimer Association. The aim of the helpline is to offer counseling about dementia, to support and guide family carers, and link them with community support services and programs. To further bolster support for dementia caregivers, there could be an expansion of existing policies to include more robust respite care options, financial subsidies, and educational programs that cater specifically to the needs of caregivers. Integrating these services into the national policy framework could alleviate some of the burdens caregivers face and provide them with the tools necessary to manage their responsibilities more effectively.

There are some limitations that should be noted; the main one is the use of the self-reported questionnaires, which is a subjective measure. The use of self-reported questionnaires can lead to potential biases, including social desirability bias and recall bias. These biases may compromise the accuracy of reported behaviors and health outcomes. Further, this is a descriptive study, limiting the power and the ecological validity of the reported findings. Comparing our results with a control group would help us to determine the extent to which caregiving itself contributes to the observed health patterns. Recognizing that non-caregivers may not experience similar struggles, makes direct comparisons challenging. Furthermore, controlling for all possible covariates that could influence the health outcomes of caregivers is a complex task. Regrettably, access to caregivers of other conditions was not feasible during our study period, which prevented us from performing a more extensive comparative analysis. Finally, the relatively small sample size of the study is another limitation. However, there are some strengths that should be noted: The study effectively highlights significant health patterns among caregivers by comprehensively evaluating their sleep, diet, and exercise habits altogether, and not one by one as previously reported. Its insights provide valuable guidance for developing targeted interventions that aim to improve the well-being of this often-overlooked group. While a portion of our sample comprised caregivers affiliated with the Athens Alzheimer’s Association, we ensured the inclusion of a more random subset, encompassing individuals without previous training or relevant information. This approach broadens the scope of our findings. Notably, the fact that many participants possessed some prior knowledge about Alzheimer’s yet still faced considerable difficulties reinforces the profound and widespread impact of caregiving for individuals with this disease. This further accentuates the necessity for comprehensive support and resources for all caregivers, regardless of their level of prior knowledge.

Continuous caregiving without adequate breaks or self-care can lead to burnout, where the caregiver feels overwhelming exhaustion, frustration, and a reduced ability to empathize with the care recipient. Maintaining good health habits helps prevent burnout by ensuring the caregiver takes time for themselves, thereby preserving their long-term capacity to care. Taking care of caregivers’ health is essential as it significantly reduces the risk of caregiver burnout, which in turn can prevent the premature institutionalization of people living with dementia. Considering these factors, it is evident that it is crucial for caregivers to prioritize their own health and well-being. By doing so, they ensure they are better equipped to handle the demands of caregiving, and also safeguard their own health over the long term. Thus, there is a need for more accessible health services, support groups and counseling, as well as increased educational resources.

## 5. Conclusions

The findings of this descriptive report paint a challenging picture of the health and well-being of caregivers, who often sacrifice their own resources to provide care. Sleep disorders, inadequate diet, and lack of exercise seem prevalent in this group, highlighting the need for targeted support systems to address these issues. Promoting self-care practices, including good sleep hygiene, balanced nutrition, and regular physical activity, could improve the health outcomes of caregivers and better enable them to fulfill their roles.

## Figures and Tables

**Figure 1 brainsci-14-00826-f001:**
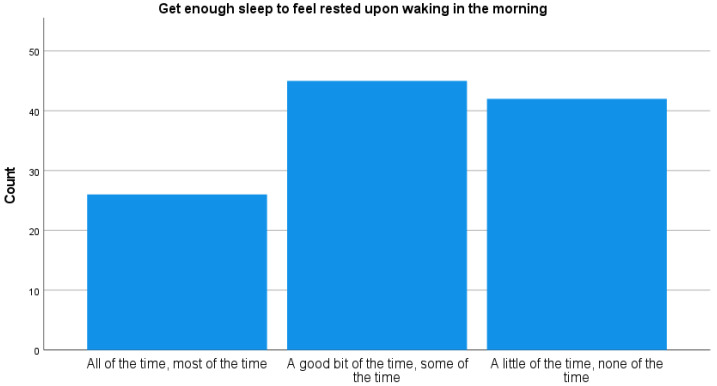
Reports on sleep characteristics.

**Figure 2 brainsci-14-00826-f002:**
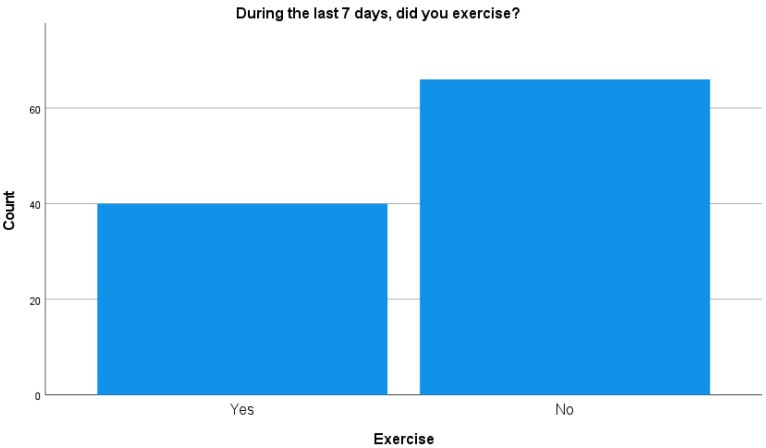
Reports on exercise habits.

**Table 1 brainsci-14-00826-t001:** Characteristics of our sample.

Characteristics	
Age, Mean (SD)	55.7 (10.4)
Sex, female, N (%)	83 (72.8)
Education, years, Mean (SD)	14.2 (5.5)
Total, N	114

## Data Availability

Data available upon request to the authors. The data are not publicly available due to privacy and ethical restrictions.

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
