# Peer review of "Sleep, Diet, and Exercise: How Much Dementia Caregivers Are Affected?"

_brainsci, 2024, doi:10.3390/brainsci14080826_

Round 1
Reviewer 1 Report
Comments and Suggestions for Authors
This brief report is a descriptive study examining sleep, diet, and exercise patterns among 114 dementia caregivers in Greece. Key findings include significant sleep dysfunction, reliance on coffee, common consumption of meat and pasta, low physical activity levels, and universal vitamin supplementation among caregivers. The study emphasizes the need for targeted interventions to improve caregivers' health and well-being
Althouhg is an interested topic, some major issues generate doubt about publication:
The study is descriptive for and small and homogeneous sample. This caregivers are in the Athens Alzheimer’s Association, wich may have worked with the carevigers in this problems. This raises concerns about the generalizability of the findings and limits the ability to draw definitive conclusions about the impact of the observed health behaviors
The reliance on self-reported questionnaires introduces potential biases such as social desirability bias and recall bias, which can affect the accuracy of the reported behaviors and health outcomes.
Without a control group of non-caregivers or caregivers in patients without dementia, it is difficult to determine the extent to which caregiving itself contributes to the observed health patterns.
Other important issues:
The study does not delve deeply into potential confounding variables (e.g., socioeconomic status, pre-existing health conditions) that could influence the caregivers’ health behaviors and outcomes.
While the background touches on relevant studies, expanding on the mechanisms by which caregiving affects sleep, diet, and exercise would enhance understanding. Also discussing how cultural factors may influence caregiving practices and health behaviors would provide additional depth. This is particularly relevant given that the study focuses on caregivers in Greece.
Provide detailed descriptions of participant recruitment, consent procedures, questionnaire administration, and data collection timeframe.
Conduct a power analysis to determine the appropriate sample size for robust statistical analysis. Why 114?
Author Response
Reviewer 1
This brief report is a descriptive study examining sleep, diet, and exercise patterns among 114 dementia caregivers in Greece. Key findings include significant sleep dysfunction, reliance on coffee, common consumption of meat and pasta, low physical activity levels, and universal vitamin supplementation among caregivers. The study emphasizes the need for targeted interventions to improve caregivers' health and well-being
Although is an interested topic, some major issues generate doubt about publication:
The study is descriptive for and small and homogeneous sample. These caregivers are in the Athens Alzheimer’s Association, which may have worked with the caregivers in these problems. This raises concerns about the generalizability of the findings and limits the ability to draw definitive conclusions about the impact of the observed health behaviors.
Thank you for your feedback. Indeed, some caregivers are members of the Athens Alzheimer’s Association; however, the sample was more diverse, including individuals without prior training or relevant information. This broader sample, while still quite homogeneous, highlights an important point. Given that many participants already had some information about dementia, the results are even more striking, underscoring the significant challenges faced by caregivers, even those who are aware of the disease. We have now included the following in the manuscript, Discussion section: “While a portion of our sample comprises of caregivers affiliated with the Athens Alzheimer’s Association, we ensured the inclusion of a more random subset, encompassing individuals without previous training or relevant information. This approach broadens the scope of our findings. Notably, the fact that many participants possessed some prior knowledge about Alzheimer's yet still faced considerable difficulties reinforces the profound and widespread impact of caregiving for individuals with this disease. This further accentuates the necessity for comprehensive support and resources for all caregivers, regardless of their level of prior knowledge.”
The reliance on self-reported questionnaires introduces potential biases such as social desirability bias and recall bias, which can affect the accuracy of the reported behaviors and health outcomes.
We now included in the Discussion section, limitations: “There are some limitations that should be noted; the main one is the use of the self-reported questionnaires, which is a subjective measure. The use of self-reported questionnaires can lead to potential biases, including social desirability bias and recall bias. These biases may compromise the accuracy of reported behaviors and health outcomes.”
Without a control group of non-caregivers or caregivers in patients without dementia, it is difficult to determine the extent to which caregiving itself contributes to the observed health patterns.
You've made a solid point about the practical difficulties in establishing an ideal control group. Indeed, a control group would definitely help us understand better the correlation between the disease and its effect on the caregivers. However, that would be difficult, since non-caregivers would not have the same struggle and we wouldn’t control for all the possible other covariates, and unfortunately, we did not have access to caregivers of other conditions. We now acknowledged that in our manuscript, Discussion section: “Comparing our results with a control group would benefit us to determine the extent to which caregiving itself contribute to the observed health patterns. Recognizing that non-caregivers may not experience similar struggles, makes direct comparisons challenging. Furthermore, controlling for all possible covariates that could influence the health outcomes of caregivers is a complex task. Regrettably, access to caregivers of other conditions was not feasible during our study period, which prevented us from performing a more extensive comparative analysis.”
Other important issues:
The study does not delve deeply into potential confounding variables (e.g., socioeconomic status, pre-existing health conditions) that could influence the caregivers’ health behaviors and outcomes.
We now added in the Discussion section the following: “When studying the health behaviors and outcomes of caregivers, it is essential to consider a range of potential confounding variables that could significantly influence the results. Socioeconomic status, for example, affects access to healthcare resources, the quality of nutrition, and overall living conditions, which in turn can impact a caregiver's ability to maintain their health while managing caregiving duties 12. Additionally, pre-existing health conditions among caregivers are critical to consider as they may predispose individuals to different stress-related illnesses or amplify the health impacts of caregiving 13,14. Although our sample was quite diverse, these factors should be incorporated into further research to control for possible factors affecting the association between the disease and its effect on the caregivers.”
While the background touches on relevant studies, expanding on the mechanisms by which caregiving affects sleep, diet, and exercise would enhance understanding. Also discussing how cultural factors may influence caregiving practices and health behaviors would provide additional depth. This is particularly relevant given that the study focuses on caregivers in Greece.
Thank you for the valuable comment. We now added the following in the Discussion section: “Caregivers often experience disrupted sleep patterns due to the nighttime needs of those they care for or from stress-induced insomnia 2. In terms of diet, caregivers may find themselves opting for more convenient, less nutritious food choices because of time constraints, leading to poorer overall health 13. Additionally, the physical and emotional exhaustion associated with caregiving responsibilities can diminish both the time available for exercise and the motivation to engage in physical activity 15. These lifestyle changes, prompted by the demands of caregiving, can significantly deteriorate a caregiver's health. Cultural factors significantly influence caregiving practices and health behaviors, particularly in a culturally distinct setting like Greece, where familial obligations and strong intergenerational support are deeply ingrained. These norms often dictate that caregiving responsibilities fall primarily to family members, influencing both the extent and nature of care provided. The traditional Mediterranean diet prevalent in Greece may generally guide caregivers towards healthier nutritional choices, yet the stress of caregiving could lead to deviations such as increased consumption of less healthy comfort foods 16. Furthermore, the cultural emphasis on familial care may result in underutilization of formal support services, increasing the physical and emotional burdens on caregivers, which can negatively impact their physical activity and sleep quality 17. Understanding these cultural nuances is crucial, suggesting that support interventions in Greece need to be culturally tailored to leverage community and familial support while enhancing access to and acceptance of external caregiving resources.”
Provide detailed descriptions of participant recruitment, consent procedures, questionnaire administration, and data collection timeframe.
We now added the following information, Methods section: “Participants were recruited from the Athens Alzheimer's Association (https://alzheimerathens.gr/en/about-us-2/). All participants were caregivers of people with all kind of dementias, of all stages. The study primarily involved familial caregivers rather than professional ones, with participants drawn from the Athens Alzheimer’s Association. These participants were either actively caring for a person with dementia or engaged with the Association's services, such as attending a dementia information seminar. Data collection was conducted solely through in-person methods, ensuring no online evaluations were carried out. Participation in the study was entirely voluntary. Data collection took place between February and March, 2024. The study was approved by the Institutional Review Committee (IRB) of the Athens Alzheimer’s Association (#4/15 February 2024). All participants have a signed consent form.
The following self-rated questionnaires were provided to the caregivers, regarding their own health:
Sleep: The Medical Outcomes Study -Sleep Scale was used 9. The scale uses predominantly Likert-type questions to evaluate sleep. Scales range from 1 (meaning “all of the time”) to 6 (“none of the time”), and require respondents to indicate how frequently during the previous 4 weeks they have experienced certain sleep-related issues. Several of these items are reverse scored.
Diet: The Food Frequency Questionnaire was used 10. This is a checklist of foods and beverages with a frequency response section for subjects to report how often each item was consumed over a specified period of time.
Exercise: The Historical Adulthood Physical Activity Questionnaire (hapaq) 11. This is an extended questionnaire regarding the daily physical activity patterns, the activity at work, and other physical recreations the person might have engaged in during the last 4 weeks of the completion time.
Questionnaires were provided in Greek. Further information regarding the questionnaires and previous use in Greek sample can be found in previous reference 12. For the purposes of the current analyses, all answers were trichotomized.
Conduct a power analysis to determine the appropriate sample size for robust statistical analysis. Why 114?
Thank you for the comment. The number was based on the amount of questionnaires we could gain in the timeline we had. This is now reflected in the Methods section as mentioned above.
Reviewer 2 Report
Comments and Suggestions for Authors
This report provides us with some data on the health conditions of dementia caregivers, which is significant for the care of dementia. However, it seems that the report is still lacking in some aspects.
- Who are these dementia caregivers? How many hours a week or a month do they spend on caregiving? Is there any financial compensation for their care?
- What policies does the country where the current research is conducted have regarding the care of dementia? It may be necessary to add some relevant policy content on the basis of the current article, exploring how the national level can provide help to dementia caregivers.
Author Response
This report provides us with some data on the health conditions of dementia caregivers, which is significant for the care of dementia. However, it seems that the report is still lacking in some aspects.
Who are these dementia caregivers? How many hours a week or a month do they spend on caregiving? Is there any financial compensation for their care?
These were part of the Athens Alzheimer’s Association. They do not spend time on caregiving, they take care of a person with dementia who might be a member of the Association. Services for the dementia people are for free. More details regarding the participants of the study (caregivers) are now provided in the manuscript, Methods section: “Participants were recruited from the Athens Alzheimer's Association (https://alzheimerathens.gr/en/about-us-2/). All participants were caregivers of people with all kind of dementias, of all stages. The study primarily involved familial caregivers rather than professional ones, with participants drawn from the Athens Alzheimer’s Association. These participants were either actively caring for a person with dementia or engaged with the Association's services, such as attending a dementia information seminar. Data collection was conducted solely through in-person methods, ensuring no online evaluations were carried out. Participation in the study was entirely voluntary. Data collection took place between February and March, 2024. The study was approved by the Institutional Review Committee (IRB) of the Athens Alzheimer’s Association (#4/15 February 2024). All participants have a signed consent form.”
What policies does the country where the current research is conducted have regarding the care of dementia? It may be necessary to add some relevant policy content on the basis of the current article, exploring how the national level can provide help to dementia caregivers.
Thank you for the valuable comment. We now added in the Discussion section the following: Greece has implemented some policies and strategies to deal with dementia and support dementia caregivers. Specifically, the National Strategy for Dementia in Greece is the National Action Plan for Dementia and Alzheimer's Disease launched by the Ministry of Health, it focuses on improving the quality of life for people with dementia and their caregivers. This includes early diagnosis, comprehensive care services and public awareness. Further, The Greek Dementia Helpline 1102 as a telephone service staffed by psychologists, neurologists and social workers with expertise in dementia care and is provided by Athens Alzheimer Association. The aim of the helpline is to offer counseling about dementia, to support and guide family carers and link them with community support services and programs. To further bolster support for dementia caregivers, there could be an expansion of existing policies to include more robust respite care options, financial subsidies, and educational programs that cater specifically to the needs of caregivers. Integrating these services into the national policy framework could alleviate some of the burdens caregivers face and provide them with the tools necessary to manage their responsibilities more effectively.

Reviewer 3 Report
Comments and Suggestions for Authors
The brief report entitled "Sleep, diet, and exercise: How much dementia caregivers are affected?" addresses a relevant topic with potential insights on how to better support caregivers of people living with dementia. This paper is generally well-written and easy to understand. The authors provide a nuanced approach to caregiving, describing it as a “challenging and rewarding endeavor,” which is pleasant to read (i.e., not only focused on negative aspects). However, the manuscript remains somewhat superficial and needs to be revised to delve deeper into the investigated topic.
The introduction is well-written, particularly the first part about sleep patterns. The authors provide a good overview of their topic and explain why it is relevant to study sleep, dietary habits, and physical activity in familial caregivers of people living with dementia. However, the sections on food intake and physical activities are less developed and contain fewer references (they need to be improved).
The methodology section needs significant improvements, especially concerning the questionnaires/scales used for sleep and diet. More information is needed about these tools (e.g., authors, number of questions, topics covered, etc.). Additionally, more details about the procedure are required (e.g., was it conducted online? Were other tests or questionnaires used?). The authors should refer to similar papers published in Brain Sciences to enhance the content of their methodology section.
The results section also needs to be reworked. The current graphics are not very useful, as the analyses are not sufficiently advanced. At the end of their introduction, the authors mentioned that their work addresses the "complex interplay of sleep, exercise, and diet needs," but this is not evident in the current analysis. It is currently only descriptive - They need to implement specific analyses to investigate these three variables together, rather than separately as is currently the case. Additionally, for participant data, the mean age alone is not sufficient; the standard deviation and the minimum/maximum ages of the participants are also needed.
In the results and discussion sections, it is difficult to put the results in perspective because there is no control group. What are the sleep characteristics of people of the same age (i.e., sleep quality typically decreases in older adults)? The same applies to food intake and physical activities. How can these changes be explained? Is it due to a lack of time? Anxiety or depression? Other factors? The paper needs more literature on these aspects and would benefit from qualitative insights.
Other remarks:
- Specify in the title or early in the text that this research focuses on familial caregivers, not professional ones.
- The authors highlight that this research question is important for the quality of caregiving. This is true, but they might also emphasize that taking care of caregivers' health reduces the risk of burnout and the premature institutionalization of people living with dementia due to caregiver burnout.
Author Response
The brief report entitled "Sleep, diet, and exercise: How much dementia caregivers are affected?" addresses a relevant topic with potential insights on how to better support caregivers of people living with dementia. This paper is generally well-written and easy to understand. The authors provide a nuanced approach to caregiving, describing it as a “challenging and rewarding endeavor,” which is pleasant to read (i.e., not only focused on negative aspects). However, the manuscript remains somewhat superficial and needs to be revised to delve deeper into the investigated topic.
The introduction is well-written, particularly the first part about sleep patterns. The authors provide a good overview of their topic and explain why it is relevant to study sleep, dietary habits, and physical activity in familial caregivers of people living with dementia. However, the sections on food intake and physical activities are less developed and contain fewer references (they need to be improved).
We now added the following regarding the questionnaires, Methods section: “The following self-rated questionnaires were provided to the caregivers, regarding their own health:
Sleep: The Medical Outcomes Study -Sleep Scale was used 9. The scale uses predominantly Likert-type questions to evaluate sleep. Scales range from 1 (meaning “all of the time”) to 6 (“none of the time”), and require respondents to indicate how frequently during the previous 4 weeks they have experienced certain sleep-related issues. Several of these items are reverse scored.
Diet: The Food Frequency Questionnaire was used 10. This is a checklist of foods and beverages with a frequency response section for subjects to report how often each item was consumed over a specified period of time.
Exercise: The Historical Adulthood Physical Activity Questionnaire (hapaq) 11. This is an extended questionnaire regarding the daily physical activity patterns, the activity at work, and other physical recreations the person might have engaged in during the last 4 weeks of the completion time.
Questionnaires were provided in Greek. Further information regarding the questionnaires and previous use in Greek sample can be found in previous reference 12. For the purposes of the current analyses, all answers were trichotomized.”
The methodology section needs significant improvements, especially concerning the questionnaires/scales used for sleep and diet. More information is needed about these tools (e.g., authors, number of questions, topics covered, etc.). Additionally, more details about the procedure are required (e.g., was it conducted online? Were other tests or questionnaires used?). The authors should refer to similar papers published in Brain Sciences to enhance the content of their methodology section.
Thank you for the comment. We now improved the Methods section as follows “Participants were recruited from the Athens Alzheimer's Association (https://alzheimerathens.gr/en/about-us-2/). All participants were caregivers of people with all kind of dementias, of all stages. The study primarily involved familial caregivers rather than professional ones, with participants drawn from the Athens Alzheimer’s Association. These participants were either actively caring for a person with dementia or engaged with the Association's services, such as attending a dementia information seminar. Data collection was conducted solely through in-person methods, ensuring no online evaluations were carried out. Participation in the study was entirely voluntary. Data collection took place between February and March, 2024. The study was approved by the Institutional Review Committee (IRB) of the Athens Alzheimer’s Association (#4/15 February 2024). All participants have a signed consent form.
The following self-rated questionnaires were provided to the caregivers, regarding their own health:
Sleep: The Medical Outcomes Study -Sleep Scale was used 9. The scale uses predominantly Likert-type questions to evaluate sleep. Scales range from 1 (meaning “all of the time”) to 6 (“none of the time”), and require respondents to indicate how frequently during the previous 4 weeks they have experienced certain sleep-related issues. Several of these items are reverse scored.
Diet: The Food Frequency Questionnaire was used 10. This is a checklist of foods and beverages with a frequency response section for subjects to report how often each item was consumed over a specified period of time.
Exercise: The Historical Adulthood Physical Activity Questionnaire (hapaq) 11. This is an extended questionnaire regarding the daily physical activity patterns, the activity at work, and other physical recreations the person might have engaged in during the last 4 weeks of the completion time.
Questionnaires were provided in Greek. Further information regarding the questionnaires and previous use in Greek sample can be found in previous reference 12. For the purposes of the current analyses, all answers were trichotomized.”
The results section also needs to be reworked. The current graphics are not very useful, as the analyses are not sufficiently advanced. At the end of their introduction, the authors mentioned that their work addresses the "complex interplay of sleep, exercise, and diet needs," but this is not evident in the current analysis. It is currently only descriptive - They need to implement specific analyses to investigate these three variables together, rather than separately as is currently the case. Additionally, for participant data, the mean age alone is not sufficient; the standard deviation and the minimum/maximum ages of the participants are also needed.
We now clarified throughout the manuscript that this is a descriptive report. We further added a descriptive characteristics table (see Table 1). SD and Mean, N was added in the Table and at the Abstract.
Table 1: Characteristics of our sample
|
Characteristics |
|
|
Age, Mean (SD) |
55.7 (10.4) |
|
Sex, female, N (%) |
83 (72.8) |
|
Education, years, Mean (SD) |
14.2 (5.5) |
|
Total, N |
114 |
In the results and discussion sections, it is difficult to put the results in perspective because there is no control group. What are the sleep characteristics of people of the same age (i.e., sleep quality typically decreases in older adults)? The same applies to food intake and physical activities. How can these changes be explained? Is it due to a lack of time? Anxiety or depression? Other factors? The paper needs more literature on these aspects and would benefit from qualitative insights.
We now re-worked throughout the manuscript. More specifically, regarding the Discussion section we added the following: “When studying the health behaviors and outcomes of caregivers, it is essential to consider a range of potential confounding variables that could significantly influence the results. Socioeconomic status, for example, affects access to healthcare resources, the quality of nutrition, and overall living conditions, which in turn can impact a caregiver's ability to maintain their health while managing caregiving duties 13. Additionally, pre-existing health conditions among caregivers are critical to consider as they may predispose individuals to different stress-related illnesses or amplify the health impacts of caregiving 14,15. Although our sample was quite diverse, these factors should be incorporated into further research to control for possible factors affecting the association between the disease and its effect on the caregivers.
Caregivers often experience disrupted sleep patterns due to the nighttime needs of those they care for or from stress-induced insomnia 2. In terms of diet, caregivers may find themselves opting for more convenient, less nutritious food choices because of time constraints, leading to poorer overall health 14. Additionally, the physical and emotional exhaustion associated with caregiving responsibilities can diminish both the time available for exercise and the motivation to engage in physical activity 16. These lifestyle changes, prompted by the demands of caregiving, can significantly deteriorate a caregiver's health. Cultural factors significantly influence caregiving practices and health behaviors, particularly in a culturally distinct setting like Greece, where familial obligations and strong intergenerational support are deeply ingrained. These norms often dictate that caregiving responsibilities fall primarily to family members, influencing both the extent and nature of care provided. The traditional Mediterranean diet prevalent in Greece may generally guide caregivers towards healthier nutritional choices, yet the stress of caregiving could lead to deviations such as increased consumption of less healthy comfort foods 17. Furthermore, the cultural emphasis on familial care may result in underutilization of formal support services, increasing the physical and emotional burdens on caregivers, which can negatively impact their physical activity and sleep quality 18. Understanding these cultural nuances is crucial, suggesting that support interventions in Greece need to be culturally tailored to leverage community and familial support while enhancing access to and acceptance of external caregiving resources.
There are some limitations that should be noted; the main one is the use of the self-reported questionnaires, which is a subjective measure. The use of self-reported questionnaires can lead to potential biases, including social desirability bias and recall bias. These biases may compromise the accuracy of reported behaviors and health outcomes. Further, this is a descriptive study, limiting the power and the ecological validity of the reported findings. Comparing our results with a control group would benefit us to determine the extent to which caregiving itself contribute to the observed health patterns. Recognizing that non-caregivers may not experience similar struggles, makes direct comparisons challenging. Furthermore, controlling for all possible covariates that could influence the health outcomes of caregivers is a complex task. Regrettably, access to caregivers of other conditions was not feasible during our study period, which prevented us from performing a more extensive comparative analysis. Finally, the relatively small sample size of the study is another limitation. However, there are some strengths that should be noted; The study effectively highlights significant health patterns among caregivers by comprehensively evaluating their sleep, diet, and exercise habits altogether, and not one by one as previously reported. Its insights provide valuable guidance for developing targeted interventions that aim to improve the well-being of this often-overlooked group. While a portion of our sample comprises of caregivers affiliated with the Athens Alzheimer’s Association, we ensured the inclusion of a more random subset, encompassing individuals without previous training or relevant information. This approach broadens the scope of our findings. Notably, the fact that many participants possessed some prior knowledge about Alzheimer's yet still faced considerable difficulties reinforces the profound and widespread impact of caregiving for individuals with this disease. This further accentuates the necessity for comprehensive support and resources for all caregivers, regardless of their level of prior knowledge.
Continuous caregiving without adequate breaks or self-care can lead to burnout, where the caregiver feels overwhelming exhaustion, frustration, and a reduced ability to empathize with the care recipient. Maintaining good health habits helps prevent burnout by ensuring the caregiver takes time for themselves, thereby preserving their long-term capacity to care. Taking care of caregivers' health is essential as it significantly reduces the risk of caregiver burnout, which in turn can prevent the premature institutionalization of people living with dementia. Considering these factors, it's evident that it is crucial for caregivers to prioritize their own health and well-being. By doing so, they ensure they are better equipped to handle the demands of caregiving, and also safeguard their own health over the long term. Thus, there is a need for more accessible health services, support groups and counselling, as well as increased educational resources.
Conclusion
The findings of this descriptive report paint a challenging picture of the health and well-being of caregivers, who often sacrifice their own resources to provide care. Sleep disorders, inadequate diet, and lack of exercise seem prevalent in this group, highlighting the need for targeted support systems to address these issues. Promoting self-care practices, including good sleep hygiene, balanced nutrition, and regular physical activity, could improve the health outcomes of caregivers and better enable them to fulfill their roles.”
Other remarks:
Specify in the title or early in the text that this research focuses on familial caregivers, not professional ones.
We now added in the Methods section: “All participants were caregivers of people with all kind of dementias, of all stages. These were mostly familial caregivers, not professional ones.”
The authors highlight that this research question is important for the quality of caregiving. This is true, but they might also emphasize that taking care of caregivers' health reduces the risk of burnout and the premature institutionalization of people living with dementia due to caregiver burnout.
We now added the following in the Discussion section: “Continuous caregiving without adequate breaks or self-care can lead to burnout, where the caregiver feels overwhelming exhaustion, frustration, and a reduced ability to empathize with the care recipient. Maintaining good health habits helps prevent burnout by ensuring the caregiver takes time for themselves, thereby preserving their long-term capacity to care. Taking care of caregivers' health is essential as it significantly reduces the risk of caregiver burnout, which in turn can prevent the premature institutionalization of people living with dementia.”

Round 2
Reviewer 1 Report
Comments and Suggestions for Authors
By enhancing the previous issues, the manuscript could be reconsidered for publication
Reviewer 2 Report
Comments and Suggestions for Authors
As a brief report, this article provides us with some data on the health As a brief report, this article provides us with valuable data on the health conditions of dementia caregivers, which is significant for the care of dementia. The authors have responded to my previous questions, and I recommend publishing this report.